# Age, Sex, and Clinical Characteristics of Juvenile Open-Angle Glaucoma Patients in a Saudi Tertiary Hospital: A Retrospective Study of Surgical and Non-Surgical Outcomes

**DOI:** 10.3390/medicina60101591

**Published:** 2024-09-28

**Authors:** Amar Almulhim, Abdulmohsen Almulhim

**Affiliations:** 1Department of Ophthalmology, Al Jabr Eye and ENT Hospital, Ministry of Health, Al-Asha 36422, Saudi Arabia; ammalmulhim@moh.gov.sa; 2Department of Ophthalmology, College of Medicine, Jouf University, Sakaka 72388, Saudi Arabia

**Keywords:** juvenile open angle glaucoma, treatment approach, outcome, age, gender

## Abstract

*Background/Objectives*: Juvenile Open Angle Glaucoma (JOAG) is a condition that presents peculiar issues because it starts at a very early age and, in the end, causes substantial vision loss. This study aimed to analyze the age and gender distribution and treatment outcomes in JOAG patients. *Methods:* We carried out a retrospective study at King Abdul Aziz University Hospital, Riyadh, Saudi Arabia, from 2015 to 2022. We extracted data from the medical records. Visual acuity data were converted to the logarithm of the minimum angle resolution (LogMAR) for standardized analysis. The CARL ZEISS Humphrey 745i Field Analyzer/HFA II-i Visual Field Analyzer was used to perform visual field examinations under the 24-2 program SITA standard. *Results:* The study involved 45 JOAG patients (87 affected eyes) with a mean age of 26.91 years. Myopia was the prevailing trait (93%), and a family history of glaucoma was found in 51.1% of cases. Most of the patients presented with severe visual field defects in both eyes (right—57.1%, left—44.4%). Regarding visual acuity, we found that the majority of affected categories belonged to either mild or moderate in both eyes. Initial and final Intraocular pressure (IOP) measurements together showed a significant reduction (*p* < 0.001) and clearly demonstrated the need for IOP control. Surgical and non-surgical treatments significantly reduced IOP, with no gender or eye differences *Conclusions:* This research offers important data concerning JOAG demographics (age and gender), clinical picture, and treatment results. Though early-onset presents challenges, multidimensional therapeutic methods have great potential to get JOAG under control and maintain visual function. Additional research is needed to study the genetic causes of JOAG and assess the long-term treatment outcomes.

## 1. Introduction

Glaucomas are a diverse group of eye conditions sharing the common feature of pathological optic neuropathy and visual field defect without (the open-angle variants) or with (the closed-angle variants) visible occlusion of the trabecular meshwork [1,2]. Primary open-angle glaucoma (POAG) is a type of glaucoma that presents with intraocular pressure (IOP) > 21, glaucomatous optic nerve changes, an open anterior chamber angle, and visual field loss in the absence of signs of secondary glaucoma and non-glaucomatous cause for the optic neuropathy [1,3]. Juvenile open-angle glaucoma (JOAG) is a rare subtype of primary open-angle glaucoma that presents between the ages of 3 and 40 years. JOAG is differentiated from early primary congenital glaucoma and secondary childhood glaucoma by normal anterior segment examination (average corneal diameter, absence of Haab’s striae, and absence of anterior segment dysgenesis) [3,4]. It is widely believed that JOAG is an autosomal dominant disease that is linked mainly with the MYOC gene and has debatable penetration among generations. Large pedigrees used for genetic linkage studies led to the identification of the MYOC gene, coding for myocilin as the most common causative gene. According to some studies, only 3.6–9.5% of JOAG patients were found to have a mutation in the MYOC gene. Affected individuals typically present with high IOP, visual field loss, and optic disc damage and are usually offered surgical management earlier. The condition poses significant diagnostic and therapeutic dilemmas due to its occurrence in individuals below the age of 40 years [5,6,7].

JOAG is a unique type of glaucoma that occurs in individuals, generally younger than 40 years old. However, JOAG, which shows early onset, poses unique diagnostic, management, and understanding challenges. Even though JOAG has some similarities with POAG, it has characteristic clinical features and demographics. It is important to understand the intricacies of JOAG for proper management of patient care and results [5,7]. The age and gender distributions of JOAG patients play a significant role in understanding the epidemiology of this condition. Additionally, it assists in the early recognition of the populations at risk and the creation of specially developed screening and diagnostic approaches [8,9]. Clinical characteristics include a large number of parameters, starting from disease presentation, visual function examination, and related ocular findings. All this makes diagnosis of JOAG rather difficult and requires personalized management techniques. Management outcomes in JOAG are multi-pronged with numerous therapeutic approaches, including drugs and surgery. One of the most crucial tasks is to assess the effectiveness of various therapeutic modalities in relation to IOP control [10,11,12]. However, the availability of data, especially from a tertiary care center in KSA, is limited in the planning of appropriate strategies. Hence, this study was conducted. The main aim of this study is to comprehensively describe the demographic profile in terms of age and gender and management results of JOAG patients. Further, we seek to contribute to the current literature on JOAG by studying the effectiveness of different therapeutic modalities in controlling IOP and answering important questions about the effects of age, gender, and laterality on IOP measurements in the JOAG population.

## 2. Materials and Methods

### 2.1. Study Design

This retrospective study was conducted at King Abdul Aziz University Hospital, Riyadh, Saudi Arabia, spanning the period between 2015 and 2022.

### 2.2. Study Population

We reviewed the medical records of patients who had been diagnosed with JOAG during the specified timeframe and had a minimum follow-up duration of at least 6 months. Both electronic and paper records were obtained for all patients diagnosed with JOAG at our tertiary referral eye care center between January 2015 and August 2022.

### 2.3. Inclusion and Exclusion Criteria

The study used the following inclusion and exclusion criteria to achieve standardization of the participant population and to remove the possible biasing factors. The inclusion criteria required that the subjects had an onset of JOAG symptoms at an age between 3 and 40 years, had glaucomatous optic neuropathy in at least one eye with visual field loss consistent with optic nerve damage, had experienced elevated intraocular pressure readings over 22 mm Hg on at least two separate occasions, and had open angles. On the other hand, exclusion criteria eliminated patients with secondary and some other forms of primary glaucoma, as well as all those with visual field defects without a connection to glaucoma and undergoing intraocular surgery. The criteria were intended to keep the study directed toward JOAG and, at the same time, preserve the accuracy of the data collected and improve the validity of the study findings.

### 2.4. Clinical Data Collection

The research adhered to the principles outlined in the Declaration of Helsinki and obtained institutional review board approval before proceeding. Data collection encompassed a wide range of clinical parameters, including systemic disease comorbidities, refractive errors, best-corrected visual acuity (initial and final), IOP measurements (initial and final), cup–disc ratio assessments, visual field assessments (mean deviation), the number of antiglaucoma medications, details of surgical interventions, the requirement for additional antiglaucoma medications postoperatively, and the occurrence of postoperative complications. A minimum follow-up period of 6 months was enforced to ensure adequate assessment of management outcomes.

### 2.5. Treatment Modality Classification

Based on their treatment modality, eyes were categorized into two groups: non-surgical and surgical. This was undertaken based on the guidelines given by the World Glaucoma Association [13].

Surgical Success Classification: Surgical success was categorized as follows.

Complete Success: Defined as achieving an IOP between 6 and 21 mmHg without the use of antiglaucoma medications and with no postoperative vision-threatening complications.Qualified Success: Similar to complete success but allowed for the use of antiglaucoma medications.Failure: Defined by the development of any of the following conditions: IOP > 21 mmHg despite maximum antiglaucoma medications, hypotony maculopathy, postoperative vision-threatening complications, or the need for another surgical intervention to control IOP.

### 2.6. Data Analysis

Visual acuity data were converted into the logarithm of the minimum angle of resolution (LogMAR) for standardized analysis. Visual field examinations were conducted using the 24-2 program SITA-standard on the CARL ZEISS Humphrey 745i Field Analyzer/HFA II-i Visual Field Analyzer (Carl Zeiss Meditec AG, Jena, Germany). Visual field defects were categorized based on mean deviation into mild (MD < −6 dB), moderate (−6 dB < MD < −12 dB), and severe (MD > −12 dB) classifications. For certain patients, particularly pediatric cases or those with poor visual acuity (worse than 20/200), visual field examination was not feasible. Refractive error (myopia or hyperopia) was assessed using a kerato-refractometer

### 2.7. Statistical Analysis

We depicted the quantitative data of the present study with mean, standard deviation (SD), and range, and the qualitative data were summarized by count and percentage. The normality of data was checked with the Shapiro–Wilk test. A *p*-value less than 0.05 was considered statistically significant. Paired t-tests were used to assess the significance of initial and final Intraocular Pressure (IOP) differences, and independent t-tests for comparing IOP values among subgroups such as right eye (OD) vs. left eye (OS), male vs. female participants, and those who underwent surgical therapy vs. non-surgical therapy.

## 3. Results

The background and baseline ocular characteristics of the studied patients are depicted in Table 1. The average age of the participants was found to be 26.91 years, with a minimum age of 6 years and a maximum age of 38 years. Fair distribution of gender was there in the cohort, with 23 males and 22 females. The analysis involved a total of 87 affected eyes distributed between the right (OD) and left (OS) eyes. Forty-two (93.3%) patients were diagnosed with bilateral JOAG, and 3 (6.7%) patients were diagnosed with unilateral JOAG. Thus, from the patient’s group representing only unilateral JOAG (3 patients), two eyes exhibited ocular hypertension, and one eye presented as healthy. The number of participants born into families with glaucomatous heritage was 23 (51.1%). The participants in the study had varied follow-up periods, with a mean of 41.62 months and a range of 6 to 84 months. Table 1 depicts the background characteristics of the studied population. Regarding baseline ocular characteristics, the majority of patients had myopia (93.3%) and presented with severe visual field defects in both eyes (OD= 57.1%, OS = 44.4%). Regarding visual acuity, we found that the majority of affected categories belonged to either mild or moderate. The mean ± standard deviation (SD) of the optic–cup disc ratio of the OD and OS were 0.73 ± 0.14 and 0.66 ± 0.17, respectively.

The surgeries performed in the surgical group were 31 (75.6%) deep sclerectomy (DS), 5 (12.5%) trabeculectomy, 2 (4.8%) ultrasound cycloplasty (UCP), 1 (2.4%) endoscopic cyclophotocoagulation (ECP), 1 (2.4%) canaloplasty, and 1 (2.4%) cyclophotocoagulation (CPC). Besides surgical failure, there were only a few complications, namely, neovascular glaucoma (1 eye—OD) secondary to a central retinal vein occlusion and hypotonia (one eye—OD and two eyes—OS), which required further interventions.

We investigated the changes in IOP by comparing initial and final measurements. Our analysis revealed a statistically significant mean difference of 17.33 mmHg between initial and final IOP values (*p* < 0.001) (Table 2).

To assess whether there were differences in IOP values between the right eye (OD) and left eye (OS), we compared initial and final IOP measurements. Our analysis indicated that there were no statistically significant differences in either final (*p* = 0.382) or initial (*p* = 0.313) IOP values between the two eyes. This suggests that in our study population, IOP values were relatively consistent between the right and left eyes (Table 3).

We explored the influence of gender on IOP values by comparing males and females. The analysis showed that there were no statistically significant gender-based differences in either final (*p* = 0.07) or initial (*p* = 0.58) IOP values. This implies that, in our JOAG cohort, gender did not appear to significantly affect IOP measurements (Table 4).

For participants who underwent surgical therapy for JOAG, we observed a mean initial IOP of 30.58 mmHg, with a standard deviation of 7.10 mmHg. After treatment, the mean final IOP decreased significantly to 14.92 mmHg (*p* < 0.01). These results suggest that surgical therapy was effective in lowering IOP in JOAG patients, highlighting its potential as a therapeutic intervention (Table 5).

The number of medications used in the non-surgical group (both OD and OS) ranged from 1 to 4 (OD [mean ± SD] = 2.11 ± 0.83; OS [mean ± SD] = 2.14 ± 0.91). Among the participants who received non-surgical therapy, the mean initial IOP was 32.97 mmHg, with a standard deviation of 6.36 mmHg. Following non-surgical treatment, the mean final IOP significantly decreased to 13.50 mmHg (*p* < 0.01). These findings indicate that non-surgical therapies were also effective in reducing IOP in JOAG patients, emphasizing their role in the management of this condition. These results collectively provide insights into the demographic characteristics and IOP-related outcomes in JOAG, shedding light on the potential benefits of both surgical and non-surgical approaches for IOP control in this patient population (Table 6).

The results indicate that the side of surgery (right or left eye) does not significantly affect the outcome. Both eyes had similar rates of complete success, qualified success, and failure. The statistical analysis confirmed that these differences were not significant (*p* = 0.806). This suggests that the surgical outcomes are likely influenced by factors other than the side of surgery (Table 7).

## 4. Discussion

This study provides insights into the age and gender distribution of JOAG in Saudi Arabia, focusing on management outcomes. This discussion will delve into the demographics of JOAG, the importance of intraocular pressure (IOP) control, specifically, age and gender-based differences, and the efficacy of various therapeutic modalities.

S. Obeidan; A. Dewedar; E. Osman A. Mousa conducted a study to report the pattern of glaucoma among Saudi patients and found that JOAG accounts for around 2% among 2296 patients presented to a Tertiary Ophthalmic University Center in Saudi Arabia, whereas an epidemiological study from USA found JOAG to represent about 4% of all childhood glaucoma [14,15]. The average age of the participants at the time of data collection in our cohort was approximately 26.91 years, with a range from 6 to 38 years. The youngest patient in this study (6 years old) is comparable with the youngest reported JOAG patient by Len V. Hua and his colleague when they reported a case of bilateral JOAG in a four-year-old Chinese boy [16]. Understanding the age distribution and gender composition in JOAG is crucial for the development of targeted screening and diagnostic strategies. Early identification of at-risk populations can lead to timely intervention and improved clinical outcomes [17,18]. In our study, a positive family history of glaucoma was seen in 51.1%, which is about two times more common than reported cases from Nigeria and Korea [15,19]. The variations in the features and prevalence of JOAG in the present study and the other two populations from Nigeria and Korea might be due to several factors such as genetic, environmental factors, and healthcare accessibility [5,20,21]. Myopia was seen in 42 patients out of 45 (93%), which is almost double what has been reported in another study [11].

The hallmark of glaucoma management is the control of IOP [22,23]. Our analysis revealed a significant reduction in IOP, with a mean difference of 17.33 mmHg between initial and final IOP measurements (*p* < 0.001). These findings align with the fundamental objective of glaucoma management—preserving visual function through IOP control. Elevated IOP is a well-known risk factor for optic nerve damage, and our results emphasize the pivotal role of IOP reduction in managing JOAG. Comparing IOP values between the right eye (OD) and left eye (OS) revealed no statistically significant differences in either final or initial IOP values. It is worth noting that asymmetrical presentation has been reported in a subset of JOAG patients in some studies [15,24]. The observed differences across the studies could be due to differences in the selection of patients, diagnostic criteria, and timing of interventions. This indicates the importance of standardized guidelines and region-specific research in understanding JOAG better.

Ng and Lung reported two cases of bilateral JOAG in a four-year-old boy and a 12-year-old girl, highlighting the importance of early detection through routine tonometry for preschool ophthalmic examination, as both patients were initially asymptomatic [24]. Timely intervention, including initial topical medication ± surgery, is crucial for preserving vision in JOAG patients. Additionally, the influence of gender on IOP was explored in our study. The analysis showed no statistically significant gender-based differences in either final or initial IOP values. However, this observation contradicts one of the earlier studies that suggested differences in IOP between males and females [19]. The potential role of gender in JOAG and its effect on IOP remains a complex and evolving area of research.

As for all glaucoma subtypes, different clinical management options have been prescribed to treat JOAG ranging from medical management to more aggressive interventions such as glaucoma drainage devices (GDD) [11,25,26]. Medical management alone is often unsuccessful in patients with very high IOP or in patients with advanced glaucoma. A cohort study from Colombia that included 36 eyes with JOAG found no surgical intervention was needed and only medical management was sufficient to control IOP, while a study from Nigeria found that surgical intervention was needed in only 28.5% (6 out of 21 cases) while 71.5% (15 out of 21 cases) were controlled with medical management alone [19,27]. Existing works of literature have reported that medical management alone failed and surgical interventions were required in the majority of cases [6,28]. Multiple surgical interventions have been reported to treat JOAG, including Selective Laser Trabeculoplasty (SLT), Gonioscopy-assisted Transluminal Trabeculotomy (GATT), Trabeculectomy with/without Mitomycin C (MMC), DS, shunt surgeries, and microinvasive glaucoma surgery (MIGS) as GATT, Kahook Dual Blade, and XEN gel implant [28,29,30]. Recently, Luo et al. proposed a fully non-invasive, non-incisional technique called femtosecond laser trabeculotomy (FLT) [31]. Initial clinical trials have demonstrated its high efficacy and safety in open-angle glaucoma patients [32], highlighting its strong potential for treating JOAG patients as well.

Our results on the impact of different therapeutic modalities in managing JOAG underscore the efficacy of surgical intervention in reducing IOP and highlight its potential as a therapeutic approach for JOAG. Non-surgical therapies also demonstrated effectiveness in our study. For patients who received non-surgical therapy, the mean initial IOP was 32.97 mmHg, with a significant reduction to 13.50 mmHg after non-surgical treatment (*p* < 0.01). This reaffirms the role of non-surgical interventions in lowering IOP in JOAG patients [27].

The present study was conducted among a specific population using IOP as the treatment outcome of JOAG. The readers of the present manuscript should consider the following limitations. Firstly, due to the study’s retrospective nature, data availability such as in a detailed analysis of optic nerve head features and Optical Coherence Tomography (OCT) was limited. Secondly, we could not evaluate the genetic predisposition (familial) inheritance. Finally, we conducted this survey in a single tertiary care center, and hence, the findings may not be similar to those of other healthcare settings and regions.

## 5. Conclusions

Our research brings important information to the growing pool of knowledge concerning JOAG since it provides data on demographics and the outcomes of different surgical and non-surgical treatments. Although progress in comprehending JOAG has been made, many issues still have not been answered, including the genetic background of gender differences and the outcomes of various therapeutic approaches. Additional studies and collaborative efforts are required to continue the process of understanding this rare and complicated condition, which will result in better care and outcomes for JOAG patients. Additional genetic studies could investigate the potential genetic etiology of this gender discrepancy, providing insights into the particular genetic determinants of JOAG.

## Figures and Tables

**Table 1 medicina-60-01591-t001:** Background and baseline ocular characteristics of the patients (*n* = 45, eyes = 87).

Parameters	Number/Mean	Proportions/Standard Deviation
Age (in years)	26.91	9.23
Gender		
Male	23	51.1%
Female	22	48.9%
Family history of glaucoma		
Yes	23	51.1%
No	22	48.9%
Follow-up period (in months)	41.62	23.02
Affected eye		
Right (OD)	42	93.3% *
Left (OS)	45	100%
Refractory status		
Myopia	42	93.3%
Hyperopia	3	6.7%
	OD	OS
Visual field defects		
Mild	2 (4.8%)	6 (13.3%)
Moderate	11 (26.2%)	12 (26.7%)
Severe	24 (57.1%)	20 (44.4%)
Visual acuity		
Normal	18 (42.9%)	23 (51.1%)
Mild	9 (21.4%)	11 (24.4%)
Moderate	10 (23.8%)	8 (17.8%)
Severe	5 (11.9%)	3 (6.7%0
Optic cup-to-disc ratio (C/D ratio) (mean ± SD)	0.73 ± 0.14	0.66 ± 0.17

* 2 eyes—Ocular hypertension and 1 eye—unaffected.

**Table 2 medicina-60-01591-t002:** Mean difference among Initial intraocular pressure (IOP) and final IOP values.

Parameter	Paired Differences					T	Df	*p* Value
	Mean	Std. Deviation (SD)	Std. Error (SE) Mean	95% Confidence Interval of the Difference				
				Lower	Upper			
Initial IOP–Final IOP	17.33	7.62757	0.80402	15.76	18.93	21.56	89	0.000

**Table 3 medicina-60-01591-t003:** Comparison of IOP values (initial and final) among OD (right) and OS (left) for 87 eyes.

Parameter	Side	*n* (Eyes)	Mean	SD	SE Mean	95% CI of the Difference		*p* Value
						Lower	Upper	
Final IOP	OD	42	14.56	4.15	0.619	−4.20	1.62	0.382
	OS	45	15.75	8.91	1.32	−4.21	1.64	
Initial IOP	OD	42	33.27	6.50	0.96	−1.40	4.33	0.313
	OS	45	31.41	7.17	1.07	−1.40	4.33	

**Table 4 medicina-60-01591-t004:** Comparison of IOP values (initial and final) among genders.

Parameter	Gender	*n*	Mean	SD	SE Mean	95% CI of the Difference		*p* Value
						Lower	Upper	
Final IOP	Male	23	12.15	3.17	0.69	−4.67	0.20	0.07
	Female	22	14.02	4.67	0.95	−4.61	0.14	
Initial IOP	Male	23	32.18	5.11	1.11	−2.87	5.02	0.58
	Female	22	31.11	7.58	1.54	−2.77	4.93	

**Table 5 medicina-60-01591-t005:** Mean values and comparison of IOP values in eyes (*n* = 41) underwent surgical therapy.

Parameter	Mean	SD	SE Mean	T	df	95% CI of the Difference		*p* Value
						Lower	Upper	
Initial IOP	30.5800	7.10013	1.00411	30.45	49	28.56	32.59	<0.01
Final IOP	14.9200	8.50412	1.20266	12.40		12.50	17.33	

**Table 6 medicina-60-01591-t006:** Mean values and comparison of IOP values in eyes (*n* = 46) underwent non-surgical therapy.

Parameter	Mean	SD	SE Mean	T	df	95% Confidence Interval of the Difference		*p* Value
						Lower	Upper	
Initial IOP	32.97	6.36	1.06	32.77	39	30.94	35.01	<0.01
Final IOP	13.5	4.26	0.67	20.11		12.18	14.9	

**Table 7 medicina-60-01591-t007:** Outcomes of Surgery Based on the Side Operated (*n* = 41).

Side of Surgery	*n*	Outcome of Surgery			Total	*p* Value
		Complete Success	Qualified Success	Failed		
Right eye	% within outcome of Surgery	57%	70%	54.5%	59.5%	0.806
Left eye	% within outcome of Surgery	43%	30%	45.5%	40.5%	

## Data Availability

The raw data supporting the conclusions of this article will be made available by the authors on request.

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
