# Peer review of "Age, Sex, and Clinical Characteristics of Juvenile Open-Angle Glaucoma Patients in a Saudi Tertiary Hospital: A Retrospective Study of Surgical and Non-Surgical Outcomes"

_medicina, 2024, doi:10.3390/medicina60101591_

Round 1

Reviewer 1 Report

Comments and Suggestions for Authors

The authors present a retrospective study on juvenile open-angle glaucoma (JOAG) patients,
examining age, gender distribution, and treatment outcomes at a single clinical center in Saudi
Arabia from 2015 to 2022. The results indicate that both medications and surgeries significantly
reduce intraocular pressure (IOP). However, other parameters, such as gender and eye
differences, were found to have no significant effect on IOP measurements or IOP reduction.
Comments:
1, In the Abstract: It would be helpful to include results for visual acuity and visual field, as
these metrics are mentioned several times throughout the manuscript, but no corresponding data
is provided. Alternatively, if these details are not the focus, it may be beneficial to reduce their
mention and elaborate further on other important aspects, such as descriptions of the devices,
methods for IOP measurement.
2, Line 261-2622: Please ensure that all abbreviations, such as SLT and MMC, are spelled out in
full the first time they are mentioned in the manuscript for clarity.
3, Line 264: For completeness in discussing current glaucoma management strategies, I would
suggest adding the following comments to the manuscript (after Line 264): Recently, Luo et al.
proposed a fully non-invasive, non-incisional technique called femtosecond laser trabeculotomy
(FLT) [1]. Initial clinical trials have demonstrated its high efficacy and safety in open-angle
glaucoma patients [2], highlighting its strong potential for treating JOAG patients as well.
References:
[1] S. Luo, T. Juhasz et al., "Evaluating the effect of pulse energy on femtosecond laser
trabeculotomy (FLT) drainage channels in human cadaver eyes", Lasers Surg. Med. 56:382–391
(2024)
[2] Nagy ZZ, Kranitz K, Ahmed IIK, De Francesco T, Mikula E, Juhasz T. First-in-human safety
study of femtosecond laser image-guided trabeculotomy for glaucoma treatment. Ophthalmol
Sci. 2023; 3(4):100313. https://doi.org/10.1016/j.xops.2023.100313

Reviewer 2 Report

Comments and Suggestions for Authors

I had the privilege to review the manuscript.
1. The title “Age and Sex Distribution of Juvenile Open-Angle Glaucoma Patients in a Saudi Tertiary Hospital” is insufficient as it omits critical aspects of the study, including information on glaucoma presentation and treatment options. A revision of the title to reflect these broader topics may improve clarity.

2. In the Results section, the description begins directly with the characteristics of patients who underwent surgery. However, it would be more appropriate to first present the demographic data before moving on to specific treatment-related outcomes. The current order is not ideal.

3. Line 142 mentions a small number of surgical complications, but it does not specify what these complications are. Providing detailed information on the types of complications would be beneficial.

4. On line 147, it is stated that “The average age of the participants was found to be 26.91 years, with a minimum age of 6 years and a maximum age of 38 years.” However, in line 216, the same data is referred to as “The average age of diagnosis.” It is unclear whether this refers to the age at the time of the diagnosis or the current age of the patients.

5. The manuscript does not provide information on the number of medications required to manage patients in the non-surgical treatment group.

6. Line 227 reports that myopia was found in 93% of patients, yet this information is not included in the Methods or Results sections.

7. The “Data analysis” section mentions that visual acuity and visual field data were evaluated, but no information on these parameters is presented in the manuscript.

8. Reporting the rates of visual field loss and optic nerve cupping at the time of glaucoma diagnosis would have been appropriate.

10. In the Conclusion, the statement that the study provides data on “the role of IOP control” is misleading. The study focuses more on the degree of IOP control achieved rather than on its role.

Round 2

Reviewer 2 Report

Comments and Suggestions for Authors

Thanks for revising the paper according to my suggestions.